# Towards a de facto Nonlinear Periodization: Extending Nonlinearity from Programming to Periodizing

**DOI:** 10.3390/sports8080110

**Published:** 2020-08-08

**Authors:** José Afonso, Filipe Manuel Clemente, João Ribeiro, Miguel Ferreira, Ricardo J. Fernandes

**Affiliations:** 1Center for Research, Education, Innovation and Intervention in Sport, Faculty of Sport, University of Porto, 4200-450 Porto, Portugal; joaoribeiro1907@hotmail.com (J.R.); ricfer@fade.up.pt (R.J.F.); 2Escola Superior Desporto e Lazer, Instituto Politécnico de Viana do Castelo, Rua Escola Industrial e Comercial de Nun’Álvares, 4900-347 Viana do Castelo, Portugal; Filipe.clemente5@gmail.com; 3Instituto de Telecomunicações, Delegação da Covilhã, 6201-001 Covilhã, Portugal; 4Superior School of Hotel and Tourism, Porto Polytechnic Institute, 4480-876 Porto, Portugal; migferreira2@gmail.com; 5Instituto Universitário da Maia (ISMAI), 4475-690 Maia, Portugal; 6Porto Biomechanics Laboratory, University of Porto, 4200-450 Porto, Portugal

**Keywords:** exercise prescription, periodization, nonlinearity, scientific definitions, programming

## Abstract

Planning is paramount in sport. Among different philosophical approaches to planning, periodization is a highly popular concept that refers to structured training periods with ensuing programs encompassing moments of progressively-loaded training, followed by recovery; it is normally deemed paramount to optimize adaptations and performance. While planning provides generic guidelines, periodization refers to the sequencing/ordering of training periods to enforce a given plan, therefore referring to longer temporal scales, and programming refers to more micro-scale aspects. In fact, similar periodization schemes may implement distinct programming strategies. Literature on the topic has used the linear and nonlinear terms to describe the content of periodized programs. However, these concepts have not been clearly defined in the literature, which may lead to inaccurate and misleading interpretations. Moreover, nonlinear periodization is usually using nonlinear programming, but with pre-stipulated sequencing of the training periods. Finally, it can be argued that nonlinearity has been an integral part of periodization since its inception, at least theoretically. In this essay, the literature was critically reviewed to better understand the validity of the linearity and nonlinearity concepts as applied in currently proposed periodization models. In addition, a novel approach for a de facto nonlinear periodization is presented.

## 1. Introduction

The human body may respond to stress in a continuum or spectrum, ranging from highly beneficial to ill-adjusted adaptations, which is the reason why a planned combination of training load and rest is required to achieve proper adaptations [1]. In this context, exercise periodization is considered determinant for properly sequencing and distributing training load (and content) across pre-established cycles [2,3,4]. In essence, this sequencing presupposes that certain orders and timings for the application of stimuli may promote better-adjusted adaptations than others; otherwise, the sequence would not have been pre-established, and the training periods would not have been stipulated in advance. There is also a role for prediction, as periodized models prescribe certain inputs (e.g., training periods and their sequencing) to achieve intended output improvements (e.g., strength and power, hypertrophy, or performance) [5,6]. This relationship between inputs and expected outputs can be described as a transformation that maps the training input to an intended adaptive output [7]. Some authors go so far as to state that periodization’s major goal is to achieve peak performance at predetermined time points [8]. As will be discussed in greater detail in a later section, however, predictability of sports performance is very limited, and contingency-necessity models may be required [9].

Generally, periodization is established in advance fitness phases and approximate timelines [10] aimed an optimal content arrangement, i.e., an optimal sequencing or ordering of themes and loads. As a side note, what constitutes an optimal load is a profoundly ill-defined concept [9]. Within each phase or period or training, different programming strategies can be used; in a sense, planning provides generic guidance, while periodization is an overarching construct for organizing temporal macro-scales, and programming would refer to the micro-planning and strategies [1]. Therefore, predetermined sequencing of training periods are common to all periodized models [11], but there is considerable plasticity in programming, i.e., in the determination of the specific loading schemes within each period [1]. In practice, some plans may be periodized, and all periodized models are implemented through specific (although changeable) programs [12]; however, not all programs are periodized (e.g., constant programs). Indeed, constant programs are a form of planning that is not periodized, concurring into a hierarchy where planning is the macro-level concept and periodization is at an optional level between general planning and programming. In this sense, we disagree with alternative conceptual models where periodization comes before planning, such as that presented by Haff [13].

The shortcomings of traditional periodized approaches started being mentioned in the 1980s, when the concept of nonlinear periodization was first introduced by Poliquin [14]. Based on a supposedly better fit to the training process requirements, nonlinear models are commonly considered more sophisticated alternatives to conventional periodized approaches [15,16]. Retroactively, former periodized models were deemed to be linear due to their gradual decreases in volume and increases in intensity. For reasons that will become clear during this manuscript, the term “linear periodization” directly contradicts the original principles of periodization [2,17]—i.e., it constitutes a lately established label that does not correspond to what periodization always aimed to be. Indeed, nonlinearity can be considered a core feature of all periodized models [18]. Despite that, self-titled “nonlinear periodized programs” have increasingly gained popularity [4,19], combining the benefits of systematization with different degrees of flexible planning and more regularly varied stimuli [14,20]. Again, we contend that flexibility was already a feature of the original periodization concepts. Nonlinear training planning templates have been appearing in various forms (e.g., nonlinear, flexible nonlinear, daily undulating, weekly undulating, block, and fractal), but nonlinear terminology remains poorly defined, technically incorrect, and potentially misleading, as it will be demonstrated.

The linear and nonlinear terms are well-defined within various scientific domains, with their distinction being fundamental to understand and model real-world behaviors [7,21]. Even so, and despite the pervasive use of linear and nonlinear terminology to describe periodized training and planning models, these terms have not been clearly defined in the periodization-related literature. Furthermore, it appears that the expression “nonlinear periodization” is actually denoting “nonlinear programming”. The objectives of the current work are therefore four-fold: (i) to question whether the current use of linear and nonlinear terminology in the training periodization literature is conceptually correct; (ii) to explore if the use of those terms provides meaningful insights, or rather serves to obscure conceptual clarity; (iii) to reinforce the assumption that optimal sequencing or ordering of training periods may not exist [9]; and (iv) to propose a novel approach to non-linear periodization, which extends nonlinearity far beyond nonlinear programming to encompass how periodization is understood.

## 2. Biological Complexity and Nonlinearity

It has become progressively evident that the behaviors of neurobiological organisms are best represented as complex and nonlinear phenomena [21,22]. In those cases, outcomes emerge from the integrated blending of multiple sub-system outputs, with the relationships between system inputs and outputs evolving over time as experiences accumulate [21,23]. Nonlinearity implies that input changes may produce disproportional output changes [7], constituting a major challenge for exercise prescription [24]. Indeed, complex adaptive systems exhibit chaotic and sensitive dependent properties, with responses to marginally differing initial conditions or noise resulting in vastly divergent emergent outcomes [22,25]. Such sensitivity implies that, barring a flawless evaluation and knowledge of all involved variables, predictions of future system states are flawed and inaccurate [21,22], requiring regular monitoring and revision [9]. Moreover, performance is multidimensional, making the qualification and quantification of load parameters a complex issue [24,26]. Overall, prediction of performance of any athlete is extremely difficult, even if the prediction is just for a few days in the future; however, some qualitative statements about the future may still be valid [9].

Therefore, these systems are expected to evolve in time, exhibiting an emergent behavior and reducing the outcomes’ prediction ability [27,28]. In this vein, linearity and nonlinearity refer to relationships between inputs and outputs, with the former meaning that a given input will always deliver the same output, and the latter denoting that an input may produce distinct outputs and that the same output may arise from different inputs [7,29]. Linearity is therefore largely independent of program design, and reflects the nature of the relationship between input and output [7,25]. In the following paragraphs, it will be questioned if the periodization-related literature accurately applies the linearity and nonlinearity concepts.

## 3. Linear vs. Nonlinear Periodization Models: A Misguided Distinction

Conventional periodization practice implicitly or explicitly incorporates timing of adaptations into its core, for reasons that will explained. Two types of timing issues may be considered relevant: the timings for achieving a certain goal or form (especially in competitive sports without weekly competitions); and the ordering of periods, i.e., the relative timings and sequencing in which each period should come about. It has long been acknowledged that human responses to prescribed interventions are nonlinear in nature [2,17], with the inherent unpredictability of human adaptation to training interventions having been widely stated [22,30,31], even if some qualitative predictions can be made [9]. The citation of Matveyev was purposeful, as it denotes that periodization has naturally advocated nonlinearity from its inception. In fact, this unpredictability is ubiquitous, and all modes of planning have to deal with this reality. Furthermore, we believe that expert knowledge and regular monitoring of training will always deliver a level of forecasting that is superior to mere guesswork. Ultimately, a distinction between linear and nonlinear periodized models would seem unreasonable, unless a claim for a linear dose–response relationship had been previously proposed—which, to our knowledge, is not the case. Periodization inevitably incorporates the concept of nonlinearity [18]. However, Poliquin [14] retrospectively coined commonly used periodized programs as being linear. In his proposal, the only difference between linear and nonlinear programs would be a more regular variation of training stimuli, including within micro- and mesocycle variation, with multiple capacities being approached during a single microcycle. Ultimately, this constitutes a misled reading of the original concept of periodization, since nonlinearity had been acknowledged from the inception of periodization [2,17]. Probably, what Poliquin meant was a wave programming with more frequent variations.

Following on the work of Poliquin [14], the application of linear vs. nonlinear terminology has become common within periodization-related literature, albeit not in a consensual way. In fact, one study refers to the use of linear and nonlinear periodized programs [32], while another one compares the effects of linear and nonlinear periodized models [33], but neither properly defined those terms. More recently, block and nonlinear periodized training interventions were compared [34], but the nature of the nonlinearity was not defined, and the reasons for using that conceptualization were not explained. In another study, linear periodized resistance training programs were applied at the beginning of each training cycle (using high volume and low intensity, and then shifting towards a reduced volume and increased intensity), while nonlinear programs involved more dramatic variations in training volume and intensity over shorter time periods [15]. According to these authors, nonlinear programs are a time-compressed form of linear proposals, ignoring the previously established qualitative difference between linear and nonlinear relationships. Furthermore, although these authors refer to nonlinear periodization, their periodization is actually established beforehand, and it is the programming that is nonlinear.

Moreover, the above-referred study used a so-called nonlinear program, with each week focusing on one capacity or skill during the first six weeks of the training plan, while the type of load was changing depending on the weekday in the following six weeks [15]. This illustrates how terms like “nonlinear”, “linear”, and “block periodization” have been used inconsistently. These authors specifically derived their nonlinear definition from the work of Poliquin [14], who misguidedly defined nonlinear programs as time-compressed forms of linear programs and further misrepresented the original concept of periodization [2,17,18].

Furthermore, we question why block periodization was not considered as nonlinear periodization in previous studies [34], and was considered as a very distinct approach. This study contrasted block and nonlinear periodization, but described that the nonlinear periodization group was practicing under three four-week blocks, and that daily content was equivalent to those used by the block periodized group [34]. Since a similar situation occurred in another study [33], it is evident that the distinction between nonlinear and block periodization is not clear.

## 4. Undulating and Flexible Periodization Models

As stated above, the nonlinear terminology has not been properly applied in the training periodization literature. However, there is a more accurate definition available, in which linearity or nonlinearity are not mentioned, and a more rigorous and appropriate terminology is used: daily undulating periodization [35], which means that the type of loads used can vary in an undulating fashion—i.e., the same training week accepts different loading schemes. Again, however, this seems to refer more to programming than to actual periodizing. Still, despite the existence of more accurate terminology, the term nonlinear became popular, and has been frequently used in the periodization literature [4,19]. A more recent study [15] cited Rhea et al. [35] to justify their approach to nonlinear periodization, but the original concept of Rhea, Ball, Phillips and Burkett [35] was that of daily undulating periodization, and not nonlinear periodization. 

Adding to the terminological confusion, another study applied the concept of flexible nonlinear programs to a strength and endurance training group, and to a maximal-effort cycling group [20]. The authors defined flexibility as the subjects’ freedom in choosing an easier workout on any given day, depending on how tired they felt, with the subjects self-adjusting their training intensity. Nonetheless, it was never mentioned why the program was nonlinear. Moreover, the flexible nature of the program was not informed by data and ongoing assessments, relying on the subjects’ personal choices. Predictions have to be regularly adjusted and changed based on current information [9], meaning that flexibility should not be equated with randomness or lack of criteria. In a non-systematic review, it was stated that flexible, nonlinear periodized programs accommodate the military training environment’s unpredictability [36]. However, the authors defined “nonlinear” as a greater variety of daily or weekly workouts, once again misusing the concept of nonlinearity. Furthermore, periodization had actually been pre-established, and it was the programming that was nonlinear. Overall, we believe the concept of flexible planning is highly valuable (whether applied to general planning, periodization, or programming), but in the referred study the criteria were not defined to inform how to implement such flexibility.

Aiming to summarize concepts and applications, it has been stated that nonlinear periodization allows further variation in workout progressions, and that flexible nonlinear periodization allows for adapting to each individual situational needs on a daily basis [16]. However, many such models were originally termed undulating, which we believe to be a better concept. Daily stimuli changes do not imply that there is nonlinear periodization or programming, as per the previously defined features of nonlinearity. Moreover, by pre-designing 8–12 week programs, [16], pre-stipulated inputs should be expected to produce outputs in a predictable manner (i.e., linearly), despite having termed the models as nonlinear. This, again, denotes that maybe the programming was flexible, but the training periods were not. Moreover, focusing on whether such loadings increase in a linear or nonlinear fashion is an unnecessary distraction, since the key guiding influence should point out that the prescribed loads are appropriate within the context of current athlete state and corresponding training objectives.

## 5. Towards de facto Nonlinear Periodization

Athletes are complex adaptive systems [37], whose training-induced responses vary considerably depending on a multitude of factors (e.g., age, genetics, and training experiences) [23,38], which is not to say that qualitative forecasting is impossible [9]. The adaptive effects of any and all training programs are inevitably nonlinear [2,31], and athletes’ developmental paths are highly individualized [39]; however, coaches’ expertise and monitoring of the training processes may provide better-quality, evolving forecasting. Predetermined prescription of training periods and their sequencing embrace a unique navigational challenge, even if their ulterior programming is flexible. In this vein, there may be an illusion of controlling the process to a greater extent than it is realistic to expect [11,31], and using previously successful maps may be inappropriate under novel conditions [40], especially because contingencies may render training plans ineffective [9]. In fact, even if planning is a necessary condition for any training process, nonlinearity should always be incorporated. Although current proposals advocate for flexible programming, the periodization itself (i.e., the sequencing of training periods and their major themes) tends to be pre-stipulated.

A contingency–necessity model may be useful here. In sports, such a model has been proposed by Sands and McNeal [9], based on a paper by Gould and several papers by Shermer. Unfortunately, we could only obtain one full paper [41]. Regardless, Sands and McNeal [9] propose an application of this model to sport. Their model recognizes the profound limitations in performing reliable predictions, especially concerning quantitative features for longer time periods. Goals and generic plans will deliver intentions, but ongoing monitoring is paramount for ensuring that such intentions are kept in check. Therefore, based on emergent information, the plans are regularly changed and updated to promote the best possible adaptations towards the intended directions. In a nutshell, this model provides strong support for nonlinearity and the importance of ongoing monitoring of training, with planning becoming a highly open process with a relevant, post hoc nature. According to the authors, periodization is an attempt to control contingencies and push performance in the intended direction, but it should evolve dynamically. In this study, a synthesis of existing planning approaches is presented in Table 1, with a novel alternative to the current periodized models, configuring a de facto nonlinear periodization also being displayed.

Most periodized models, even the so-called nonlinear programs (and recognizing that nonlinear periodization is commonly being used to actually denote nonlinear programming), start by determining key dates (e.g., main competition or events) to establish the endpoints, and then proceed with backward establishment of training periods and their sequencing, starting with the last weeks of the season and regressing until the first week of training [2]. Although longer competitive seasons have somewhat downgraded the notion of timings for achieving peak form, the sequencing or ordering of the training periods is still a major part of periodized models. Indeed, what all periodized processes have in common is the existence of predetermined contents for each training cycle (although the specifics vary from model to model, and the programming might be nonlinear) and predetermined sequencing or ordering of those training periods. Therefore, such approaches assume a linear relationship between load and adaptation in terms of temporal macro-scales, although incorporating nonlinearity into the programming (i.e., micro-scales). In comparison, non-periodized constant programs have a single training period and lack novelty (again, they are examples of programming without periodization), while non-periodized random processes have no coherent structure at all.

Alternatively, a possible solution to conjugate structured planning with non-linear dynamics may be a varied but not a priori periodized strategy (i.e., not merely limited to nonlinear programming). Here, short-term programming would be coupled with ongoing methods for controlling the training process, but the sequencing or ordering of the training periods would not be pre-stipulated. Despite being guided by overarching planning goals, this approach would be highly open to changes derived from emerging information. While current periodized models largely predetermine the sequencing of training periods (even if their duration may be changeable and their programming nonlinear), our proposal takes nonlinearity one step further, and proposes that the sequencing itself is not pre-stipulated—i.e., training periods and the subsequent periodization would constitute an emergent feature of the process. Moreover, emerging information would result from the processes established for training control and monitoring, instead of representing mere unsubstantiated or random choices [9]. In fact, this model should not be equated with absence of planning, as all training processes require proper planning. Our proposal is to plan overarching themes and goals, and then use short-term programming coupled with ongoing training control, with the training periods becoming an emergent property and not an a priori definition. Therefore, there is still intentionality in the process, but the plan becomes much more accepting of relevant incoming information, using it to its favor in the establishment of the specific training plans. From this perspective, nonlinearity would not be limited to programming, but would be extended to periodization as well.

## 6. Concluding Remarks

Planning is an essential feature of life, including sports, and such plans may or may not be periodized. For reasons that have been previously explained, we depart from definitions that consider periodization to be above the level of planning [13]. The current work presupposes that load is nonlinearly related with adaptation, and that current periodized models (even those using so-called nonlinear programs) do not respect such relationships, thereby misapplying the concepts of linearity and nonlinearity. As was stated, the so-called nonlinear periodization models either do not define what the term “nonlinear” means, or use definitions that are in sharp contrast with well-established scientific definitions. Furthermore, although previous authors’ programming was nonlinear and flexible, their periodizing has been largely predetermined, i.e., the ordering or sequencing of the different training periods were pre-stipulated well in advance. The main issue here is that the majority of the researchers have been applying the term “nonlinear” as if it was different from a linear approach in terms of degree, while in reality the two differ qualitatively, i.e., they are dichotomic entities with very distinct behaviors [7,21]. Moreover, nonlinearity has been a core tenet of periodization since its inception [2], despite what Poliquin [14] suggested. However, it is a fact that periodization models tend to assume certain cause-and-effect relationships between training stimuli and the subsequent outcomes [9]. Nonetheless, it is our conviction that this theoretical problem is not affecting high-level coaches’ practice, as they constantly monitor, control, and adjust the process, thereby regularly introducing changes to the plan, including the sequencing of training periods [42].

However, such scientific misconceptions may negatively impact the design of training sessions grounded on specific linear or nonlinear thinking models. Furthermore, the trending appeal for nonlinearity may cast a shadow over so-called linear periodization models, whose merits may be unjustly diminished (and, in fact, as was demonstrated, were not actually attempting to be linear). If nonlinearity is to be applied, analytical approaches will not suffice, and predictions will often be limited to generalities; even so, they will most likely fail in the long-term [21]. Qualitative predictions concerning performance may be relatively successful in terms of the direction of the adaptations, but they are less successful in determining the magnitude of such adaptations [9]. In this vein, we have put forth a proposal for a de facto, nonlinear periodized approach, consisting of generic planning implemented through micro-level flexible programming coupled with ongoing training control, while the actual training periods and their sequencing would constitute an emergent feature of the process. This extends the concept of nonlinearity from programming to the more overarching concept of periodizing, by not attempting to pre-establish the sequencing or ordering of training periods. An additional, more mathematically-driven account of linearity is provided in the Appendix A.

## 7. Practical Implications

The literature on exercise periodization is misapplying the concepts of linear and nonlinear, expecting a linear relationship between input and output. We recommend that the terms linear and nonlinear are replaced by more appropriate terms (e.g., “undulating periodization”). Furthermore, regardless of their specificity, periodized programs should be flexible, and it is our conviction that coaches already interpret training programs in such a way [42], therefore making the term “flexible” somewhat redundant. For a de facto nonlinearity to be attached to the concept of periodization, nonlinear thinking has to be incorporated not only to the micro-scale of programming, but also to the more macro-scale of periodizing. Research should focus on the establishment of ongoing control processes, so that flexible planning is not equated with random decision-making.

## Figures and Tables

**Table 1 sports-08-00110-t001:** Different approaches to training periodization.

Input–Output Relationship	Type	First Training Cycle Contents and Duration	*n*th Training Cycle Contents and Duration
Linear	Periodized (a priori): includes all current existing periodized approaches, including so-called nonlinear approaches. The sequencing or ordering of the training periods are pre-stipulated. Depending on the model, programming can be flexible.	Predetermined contents (e.g., *A*) and duration is estimated (e.g., *X* weeks). Depending on the model, programming can be flexible, and duration can be changed.	Predetermined contents and duration are estimated. Depending on the model, programming can be flexible, and duration can be changed.
		Example of a predetermined sequence: The first cycle would focus on general motor coordination (±3 weeks), the second cycle on aerobic conditioning (±4 weeks), the third cycle on muscle hypertrophy (±6 weeks), and so forth, until the *n*th cycle, which could focus on maximal strength (±3 weeks).	
	Non-periodized constant	The program is constant, but loads can be progressively incremented. Example: resistance training program using three sets of 12 repetitions. Load can be incremented when the subjects can perform sets of 15 repetitions. The exercises and their order do not change during the program.	
	Non-periodized random	There is no overarching plan and changes are random. We believe that there should be no example of this in sports training. However, we do know applications of it in the context of personal training, e.g., the deck of cards strategy, where each day the client picks a card at random and performs the training included in that card. We do not advocate this strategy.	
Nonlinear	Non-periodized but varied (non-random). Training periods become emergent features, i.e., the sequencing is not pre-stipulated.	Predetermined contents (e.g., *A*) and undetermined duration. Flexible programming.	Undetermined. Stipulated near the end of the previous training cycle. Flexible programming.
		Example: first training cycle devoted to improving skills. The first cycle ends only when technical form has achieved the intended quality. Near the end of the first cycle, depending of the responses of the athletes, decide whether the second cycle should focus on conditioning and technical applications under fatigue, or instead develop power and speed. Therefore, the sequencing and duration of training periods is not pre-defined.	

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
