# Peer review of "Towards a de facto Nonlinear Periodization: Extending Nonlinearity from Programming to Periodizing"

_sports, 2020, doi:10.3390/sports8080110_

Round 1
Reviewer 1 Report
Congratulations your article as bright as necessary.
I am really shocked at the clarity of the exposition and I fully share your suggestions.
I have nothing further to add only I have really enjoyed reading
Author Response
- The reviewer has made some highly positive comments on the work, for which we thank him/her. There were no recommendations for changes.
- Therefore, we thank the reviewer for endorsing this work and for the very kind words.
Reviewer 2 Report
See attached document.

Reviewer 3 Report
This is an insightful and novel paper that addresses the better understand the validity of the linearity and nonlinearity concepts as applied in currently proposed periodization models.
Most previous studies, considering periodization models relate to its effectiveness, however, these concepts have not been clearly defined, which may lead to inaccurate and misleading interpretations. The authors did a good job addressing the numerous shortcomings of previous research.
Besides, new valuable scientific data, the paper includes interesting, and significant practical implications for coaches, recreationally active people, and competitive athletes.
I enjoy reading this manuscript and I have only one technical comment related to table 1. I believe that the information contained in the columns could be presented in a more concise way, from third to sixth columns provided information are the same. I suggest the authors to reconsider the designing of table 1, maybe authors could provide a brief example of those training periodization approaches?.
Overall, in my opinion, the paper is suitable for publication in Sports MDPI.
Author Response
- Reviewer 3 delivered very positive comments on the manuscript and endorsed the work done, similarly to reviewer 1.
- However, reviewer 3 suggested reducing some redundancy presented in the table, specifically in columns 3 to 6. We have eliminated columns 4 and 5, as we believe columns 3 and 6 were sufficient to provide the idea we were trying to get across. Therefore, the table now lost redundancy.
- Furthermore, reviewer 3 suggested the inclusion of a brief example of those training periodization approaches. In the table, each row now includes one example to illustrate the presented models.
- We thank reviewer 3 for the very kind words and for endorsing this work.
Round 2
Reviewer 2 Report
I appreciate the authors' willingness to discuss my comments from the initial round of reviews. I agree with the authors - that critical reflection and consideration of my comments and those of other reviewers have improved the manuscript. Though we do not necessarily agree on each point, I do believe that the authors have presented sound scientific arguments and reasonable interpretations of the previous literature in constructing their arguments. I believe that this will be a well-received publication on the topic of periodization.